# Synthesis of a Compound Phosphorus-Nitrogen Intumescent Flame Retardant for Applications to Raw Lacquer

**DOI:** 10.3390/polym13172858

**Published:** 2021-08-25

**Authors:** Bing-Chiuan Shiu, Kunlin Wu, Ching-Wen Lou, Qi Lin, Jia-Horng Lin

**Affiliations:** 1Fujian Engineering Research Center of New Chinese lacquer Material, College of Material and Chemical Engineering, Minjiang University, Fuzhou 350108, China; bcshiu@mju.edu.cn (B.-C.S.); kunlinwu2020@163.com (K.W.); cwlou@asia.edu.tw (C.-W.L.); 2Laboratory of Fiber Application and Manufacturing, Department of Fiber and Composite Materials, Feng Chia University, Taichung 40724, Taiwan

**Keywords:** raw lacquer (RL), inflaming retarding, intumescent flame retardant, combustion performance, heat stability

## Abstract

Raw lacquer (RL) is a natural polymer compound with highly promising applications; however, its inflammable attribute restricts the industrial applications. In this study, melamine is used to formulate tri (1-melamine-2-propanol) phosphate (FR-1), after which it is synthesized with ammonium phosphate (FR-2) and diatomite to form a compound phosphorus-nitrogen intumescent flame retardant (IFR). Next, IFR is used as the filling agent that then cross-links with RL, and as such RL/IFR membranes are formed after the curing. The limiting oxygen index (LOI) measurement, the vertical combustion test (UL-94), the microshape calorimetric analysis (CCT), and the thermal gravimetric analysis (TGA) are conducted to examine the combustion resistance and thermal stability of the membranes. Fourier transform infrared spectroscopy (FT-IR) and electron scanning microscope (SEM) are performed to separately characterize the structure and compatibility; the mechanical properties of the membranes are also evaluated. The vertical combustion test results confirm that with 30 wt% of IFR, RL/IFR membranes acquire 12.3% higher LOI and a vertically combustion of V-0 level. The TGA indicates that RL/IFR membranes demonstrate a greater adhesion level, a higher rigidity, and better luster than pure RL membranes.

## 1. Introduction

As one of the oldest paints used in northeast Asian countries, such as China, Japan, and Korea, raw lacquer dates back more than seven thousand years ago. Raw lacquer is a secretion from Toxicodendron vernicifluum, featuring eco-friendly regeneration [1]. It primarily contains 60~65% of urushiol, 20~30% of water, 5~7% of gummy substance, 2% of glycoproteins, and 0.2% of laccase [2,3]. The main component, urushiol, is composed of catechol derivatives with different unsaturated carbon chains that consist of 15–17 carbon atoms as Figure 1 [4]. Via the accelerating urushiol, laccase undergoes coupling reaction and autoxidation reaction over the long aliphatic unsaturated side chain, which in turn leads to the curing of raw lacquer films [5,6]. The unique structure of urushiol provides raw lacquer films with luster, a high adhesion of matrices, good thermal stability, excellent acid resistance, durability, and antibacterial properties [7,8], which is why raw lacquer has been used for the decoration of furniture and handcrafts since several centuries ago [4]. Over the process of time, raw lacquer is now used for more versatile applications, such as ships, submarines, and the maintenance of underwater infrastructure [4]. However, raw lacquer also has some disadvantages. For example, raw lacquer is extremely inflammable, highly viscous, and demands a long curing time. Additionally, it lacks ease of processing and alkali resistance, and these disadvantages restrict its industrial application to a considerable extent, especially the inflammable attribute, which is a hidden danger to the occurrence of fire. Therefore, raw lacquer needs to be reinforced with combustion resistance in order to reduce the loss of the finance and ensuring the public safety, which is significant to broaden the use of raw lacquer.

There are worldwide studies investigating raw lacquer. Je et al. conducted the curing reaction between natural urushiol and Fe^3+^. The cross-linked network urushi that is formed by chemical bonds of urushiol and Fe^3+^-coordination bonds was used as an underwater adhesive [9]. Lu et al. wrapped Ag nano particles in natural urushiol to form ocean anti-fouling coating, investing how the coating layer constrained the growth of fungus and microalgae [10]. Lone et al. synthesized an environmentally protective, highly efficient antibacterial/anti-fouling coating that was composed of natural urushiol. Moreover, a great deal of research focused on the disadvantages of raw lacquer, trying to modify raw lacquer [11]. Xia et al. employed UV lights to solidify raw lacquer films and successfully shortened the curing time [12]. Yang et al. used copper modified montmorillonite (Cu(II)-MMT) as the highly efficient catalysis and proved that it could accelerate the curing time of urushiol, improving the luster and thermal stability concurrently [13]. Gao et al. modified raw lacquer with hexamethylenetetramine (HMTA), during the thermal curing condition HMTA was dissociated into amine and formaldehyde, the latter of which underwent the condensation reaction along with urushiol [14]. As a result, the curing time was significantly decreased with the luster and alkali resistance being strengthened. Zhang et al. incorporated different contents of dendritic polyamidoamine (PAMAM) with raw lacquer, examining the effects of PAMAM over the curing process and properties of raw lacquer films. The test results showed that the comprehensive performances of raw lacquer films were improved [15]. In addition, the cross-linking between PAMAM and urushiol also helped reduce the curing time and strengthen the alkali resistance for raw lacquer.

The more the studies on lacquer, the more widespread the applications of raw lacquer. Raw lacquer is used in furniture, artifact, decoration, construction, and corrosion resistance, which increasingly urges the combustion resistance of raw lacquer. Nonetheless, there are few studies on the modification of flame-retardant raw lacquer. Flame retardants can help with the combustion resistance of polymer materials. Based on the different uses, flame retardants can be divided into reactive flame retardants and added types of flame retardants, the latter of which reacts with polymers exclusively at high temperatures. Therefore, the added types of flame retardants show ease of processing and have a low cost. Considered to have good potential, intumescent flame retardants possess a dehydrating agent, foaming agent, and carbonization agent concurrently with advantages such as good thermal stability, weatherability, a low amount of smoke, and low toxicity. Subsequently, intumescent flame retardants have received widespread attention from scholars globally. 

During the combustion process, phosphorus compounds of flame retardants are heated to dissociate into strong acids with either an absorbent effect or dehydrating effect. The product is mainly to expedite the dehydration carbonization of polyhydroxy compounds, thereby forming a char layer that is flame retardant and able to isolate the polymers, oxidants, and the heat sources. As a result, the substance and heat quantity are blocked from transmission, which in turn stops the combustion [16]. With the help of high temperatures, the foaming agent of flame retardants is decomposed to generate flame retardant or flame-resistant airs that can dilute inflammable air from the surrounding mixed air, while reducing the oxygen content of the mixed air. Subsequently, an air protective layer is formed, surrounding the flammable materials and with the ability to consume considerable heat quantity [17]. In the meanwhile, the flame retardants are thermally melted, generating a highly viscous liquid that covers the surface of flammable materials while blocking the external heat source from the transmitting the heat and oxygen into the material’s interior and at the same time preventing inflammable air from being emitted from the flammable materials [17,18,19].

Besides, silica particles are used as the filler of polymer material in order to reduce the material cost. More studies have revealed that silica particles exhibit good combustion resistance when used as the filler [20]. Silica-contained flame retardants have garnered more and more attention due to their extraordinary combustion resistance, good mechanical properties, high fluidity, good heat resistance, low smoke emission, and low toxicity. In this study, tri (1-melamine-2-propanol) phosphate (FR-1) is first synthesized and then combined with ammonium phosphate (FR-2) and diatomite at different ratios to form a new compound, phosphorus-nitrogen intumescent flame retardant (IFR), that is used as a highly efficient flame retardant for raw lacquer (RL). The combination of IFR and RL forms combustion resistant RL/IFR membranes. Afterwards, the thermal stability, combustion resistance, and optics properties of the membranes are evaluated, examining the combustion resistance mechanism. 

## 2. Experimental

### 2.1. Materials 

Chinese lacquer was purchased from Sichuang and was filtered using gauze in advance. Melamine was purchased from Chinasum Specialty Products Co., Ltd. (Jiangsu, China). Epichlorohydin, ethenediamide, sulfuric acid, sodium hydroxide, and ethanol absolute were purchased from Fuchen Chemical Reagents Factory, Tianjin, China. Phosphoric acid and potassium phosphate trihydrate were purchased from Zhanyun Chemical Co., Ltd, Shanghai, China. Diatomite was purchased from Alighting Chemical Co., Ltd., Shanghai, China. 

### 2.2. Characterizations

The vertical burning rate is based on the test standards GB/T2408-2008, UL94-2009, the sample size is 120 × 50 × 0.075 mm^3^, and the vertical burning instrument YG(B)815D-I instrument (Wenzhou Darong Textile Instrument Co., Ltd., Wenzhou, China), the results were classified with a burning rating of V-0, V-1 or V-2.

The LOI test standard is based on GB/T 2406-80 and ASTMD 2863-97. The sample size is 120 × 50 × 0.075 mm^3^, which was measured using a F101D instrument produced by (Qianshi Precision Electromechanical Technology Co., Ltd., Shanghai, China).

The impact strength was measured using QCJ-50/100 paint film impactor (Tianjin World Expo Weiye Glass Instrument Co., Ltd., Tianjin, China). The film with the biggest shock distance was recorded as film impact resistance values. 

The scratch hardness of the film was measured with a pencil hard-ness testing (QHQ-A, Weichuangjie Testing Instrument Co., Ltd., Shenzhen China), which is one of the methods used for the physical characterization of polymer films. The hardness from the highest to the lowest is expressed as 6H, 5H, 4H, 3H, 2H, H, F, HB, B, 2B, 3B, 4B, 5B, and 6B based on the pencil hardness.

The IR spectra of the membranes were collected by the ATR method using an FT-IR spectrometer (MPIR8400S, Shimadzu, Japan). Thirty-two scans were conducted with a resolution of 4 cm^−1^.

TGA experiments were carried out with a STA449F3 thermal analyzer from 30 °C to 800 °C with a heating rate of 10 °C/min. SEM images were analyzed using the field emission SEM (Hitachi S3400, Japan) at an acceleration voltage of 2 kV.

### 2.3. Synthesis of FR-1 and FR-2

#### 2.3.1. Synthesis of FR-1

FR-1 was synthesized according to Scheme 1. Synthesis of FR-1 was carried out in a dry three-neck flask equipped with a stirrer and an oil bath. Melamine (4.1259 g), epichlorohydrin (2.6 mL), and deionized water were added to the flask. Next, 2.9458 g potassium phosphate trihydrate is dissolved in deionized water, and the resulting solution (25 mL) was then added when the mixture was refluxed with stirring. After the mixture turned clear, it was refluxed for 0.5 h again. After cooling to room temperature, the reaction mixture was filtered under a low pressure, thereby obtaining the solid. Finally, the solid was rinsed by deionized water and vacuum dried at 120 °C for 4 h. 6.13 g of white solid was obtained and designated as FR-1 (yield: 89%, The yield refers to Formula (1)). FTIR (KBr cm^−1^): 1025.98 (P-O-C), 3334.88 and 3133.26 (-NH and -NH_2_), 3469.13 (ROH), 1198.43 (P=O), 814.04 (melamine ring).

#### 2.3.2. Synthesis of FR-2

Synthesis of FR-2 was carried out in a dry three-neck flask equipped with a stirrer and cooling water bath. Ethenediamide (10 mL) and ethanol absolute were directly introduced into the flask. Next, a desired amount of phosphate acid was added until the mixture became weak acid. Then, the mixture was filtered under a low pressure and rinsed with an ethanol absolution. After being vacuum dried at 100 °C for 4 h, 18.06 g of white solid was obtained and designated as FR-2 (yield: 97%, The yield refers to Formula (1)).
H_2_NCH_2_CH_2_NH_2_ + H_3_PO_4_→[H_3_NCH_2_CH_2_NH_3_]_3_[PO_4_]_2_
(1)$$\mathrm{yield}=\frac{\mathrm{active}\;\mathrm{output}}{\mathrm{theoretical}\;\mathrm{yield}}*100\%.$$

### 2.4. Preparation of RL/IFR Membranes

With different ratios, FR-l, FR-2, diatomite, and raw lacquer were made into diverse RL/IFR membranes. According to the test results, the optimal combustion resistance occurs when samples consisted of FR-1, FR-2, and with diatomite at a ratio of 56 wt%, 16 wt%, and 28 wt%. Next, the intumescent flame retardant is added to the filtered raw lacquer and then blended for five minutes at room temperature, forming RL/IFR mixtures. According to the Chinese standard (GB/T 1727-1992), RL/IFR mixtures are smeared over a glass slide and a tinplate and kept at 26 °C and a relative humidity of 80% to dry naturally. The RL/IFR membranes are trimmed into 120 × 50 × 0.075 mm for the vertical combustion test. In order to optimize the combustion resistance, the IFR content is changed based on the following Formula (2) computation where *w*1 and *w*2 means the content of IFR and RL, respectively.
(2)$$\mathrm{Percentage}\;\mathrm{ratio}\;\left(\%\right)=\frac{w1}{w1+w2}\;\times \;100\%.$$

## 3. Results and Discussion

### 3.1. FT-IR Spectrum Analysis 

Figure 2 shows the FT-IR spectra of RL, RL/IFR, and IFR. As for the spectra of IFR, the wider characteristic peak at 3300~3500 cm^−1^ is a result of the overlapped -OH and -NH- absorption peaks. In addition, –NH_2_ absorption peaks are presented at 3133.26 cm^−1^ and the stretching vibration of Melamine rings is presented at both 1198.43 cm^−1^ and 814.04 cm^−1^. The C–N stretching vibrations of melamine are identified between 1652 and 1440 cm^−1^ and the characteristic peaks of FR-1 and FR-2 occur at 1256 cm^−1^ (P=O), 1075 cm^−1^ (P-O symmetrical expansion), and 880 cm^−1^ (P-O asymmetric expansion) [21].

As for RL’s spectra, the absorption peaks of urushiol’s C-O-H group occur at 3439, 1363, 1275, and 1150 cm^−1^, which correspond to the stretch vibration of O-H, β O-H, γ O-H, and C-O, respectively [13,21,22]. Comparing FT-IR spectra of RL and RL/IFR, it is distinctive that RL and RL/IFR have comparable absorption peaks with slightly different strength and width. However, the presence of the asymmetric stretch of P-O at 880 cm^−1^ indicates that raw lacquer covers the absorption peak of IFR. 

### 3.2. Thermal Stability of Flame Retardant RL/IFR Membranes

The flame retardant RL/IFR membranes are measured with a thermal gravimetric test to study the thermal stability and decomposition behavior. The TG and DTG curves are shown in Figure 3. There are three decomposition stages of RL/IFR membranes. The first stage occurs at 30–248 °C where RL and RL/10% IFR membranes separately have a weight loss of 6.2% and 4.7% [23], which is caused by the decomposition of oligomer from raw lacquer as well as the moisture loss of membranes. The second stage occurs at 248–400 °C, which is ascribed to the decomposition of polysaccharides and glycoprotein [21]. Meanwhile, the weight loss rate is 30.2% for RL membranes and 29.9% for RL/10% IFR membranes. The stage was very similar to the stages of some polysaccharides reported in [3,24]. In the third stage (400–500 °C), the weight loss rate is 84.3% for RL membranes, which is due to the decomposition of urushiol polymer [21]. At the same time, the weight loss rate for RL/10% IFR membranes is 72.5%, which is highly correlated with the thermal decomposition of IFR’s phosphorous compounds.

RL membranes exhibit two major decomposition stages based on the DTG curves, and the maximal thermal decomposition degree of T_max1_ and T_max2_ is individually 309.1 °C and 444.6 °C, which can be attributed to the decomposition of the lacquer polysaccharide and glycoproteins as well as the degradation of urushiol polymers [3,25]. Moreover, according to the research, with the increase of the content of expansion flame retardant, the mechanical properties of the film decreased, the maximal decomposition degree (T_max1_ and T_max2_) of RL/10% IFR is individually 309.4 °C and 455.5 °C, which indicates that the presence of IFR does not attenuate the decomposition of the lacquer polysaccharide and glycoproteins. In fact, most of the phosphorus or nitrogen containing flame retardants may cause the decomposition initiation in advance of the composites because they should display a gas or condensed phase flame retardancy action before the material degradation [26]. As a result of the high temperature, the strong acid released by IFR can accelerate the dehydration carbonization of urushiol, reducing the degradation of urushiol polymer and enhancing the thermal stability of RL/IFR membranes.

### 3.3. Combustion Analysis of Flame Retardant RL/IFR Membranes

Figure 4 shows that RL/IFR membranes are examined by the vertical combustion test (UL-94). The RL membranes are on fire drastically when set alight, and the non-stop fire is then accompanied droplets. With 10 wt% IFR being incorporated, the RL/IFR membranes exhibit a weak flame after being set alight. Afterwards, a thick char layer is generated, which in turn blocks the polymer from contacting the flame and oxygen, and eventually stops the heat transmission. With 20 wt% IFR, the RL/IFR membranes can be set alight but without constantly burning. Namely, the flame is efficiently put out due to self-extinguishing and the membranes will not be set alight once again. Finally, with 30 wt% IFR, the RL/IFR membranes cannot be set alight at all, indicating an excellent combustion resistance. Because the compound phosphorus-nitrogen intumescent flame retardant (IFR) contains a charring agent, a dehydrating agent, and a foaming agent, the foaming agent dissociates the flame retardant gas at high temperatures, diluting the concentration of flammable air from the surrounding mixed air while reducing the oxygen content concurrently. As a result, the foaming agent can form an air protective layer while removing considerable heat energy. The dehydrating agent triggers dehydration and carbonization of the polyol, which in turn forms a char layer that has a certain thickness and is not easily burned, thereby isolating the materials, oxidants, and the heat source to prevent the transmission of material and heat energy. When the diatomite is heated, the low surface energy of silicon allows the silicon to move over the polymer surface, forming a protective layer that resists high temperatures and prevents its underneath layer from being thermally decomposed. 

RL/IFR membranes are tested using the burning test, examining the influence of the IFR content on the residual carbon rate (Table 1) and the intumescent effect (Figure 5). The solidified RL/IFR membranes are trimmed into an identical size and then burned at 500 °C for 1.5 h in a muffle furnace. Figure 5a shows that RL membranes are totally burned into ashes at 500 °C with a residual char rate being 0%, and Figure 5b shows that with 10 wt% IFR, the RL/IFR membranes generate a rather fluffy char layer with a residual char rate being 13. 98%. By contrast, with 20% of IFR, the char layer has an improved density with a residual char rate that is 30.76% higher than that of RL/10% IFR membrane. The residual char rate of RL/30% IFR is 8.12% higher when compared to that of RL/20% IFR membranes, and Figure 5e also shows that RL/30% IFR membranes exhibit the maximal intumescent effect. To sum up, the residual char rate first increases and then decreases with a rise in the IFR content. Containing a low IFR content, phosphorus compounds first dissociate and generate strong acid in response to the heat. The strong acid effectively accelerates the dehydration and carbonization of the polyol. Afterwards, with a constantly increasing IFR content, the strong acid only exerts a positive influence over the dehydration and carbonization of the polyol to an insignificant extent based on the mechanism where (C_6_H_10_O_5_)n→6nC + 5nH_2_O. 

Figure 6 shows the SEM images of the surface and cross section of a char layer. Based on Figure 6a, the char layer of RL/10% IFRL appears smoother, but there are many chambers in the interior. Because IFR emits both flame resistant air and inflammable air in the form of chambers inside the char layer, which subsequently increases the thickness of the char layer and thus prevents the heat energy and flame from spreading, the RL/IFR membranes demonstrate better combustion resistance. Besides, over the chambers in the cross section of the char layer are many spots. Figure 6b shows the presence of many pores over the char layer and many chambers over the cross section. The results are ascribed to the vapor, N_2_ and CO_2_ (i.e., flame resistant air types) that are produced during the thermal decomposition of IFR, and then sealed in a thick char layer, which subsequently making the char layer in an intumescent state. In the meanwhile, a tremendous amount of gaseous products are released from the pores over the char layer, which in turn puts out the fire efficiently, known as the blow-out effect [22]. At last, with 30 wt% IFR, there is a greater flame retardant content for the RL/IFR membranes, facilitating the formation of chambers over the char layer surface, as shown in Figure 6c. In conclusion, a rise in the phosphorus compound has a positive influence on the formation of a dense char layer [26,27,28]. 

### 3.4. Micro-Scale Combustion Calorimetry (MCC)

To further examine the flame retardant effect as related to the IFR content, MCC is used to evaluate the RL/IFR membranes, examining the related parameters and mechanism during the combustion. Table 2 displays the corresponding parameters: heat release rate (HRR), peak of heat release rate (PHRR), total heat release (THR), and ignition temperature. The incorporation of IFR effectively reduce the aforementioned parameters. RL/10%IFR membranes have a HRC of 175 J·g^−1^·K^−1^, PHRR of 174.9 w/g, and THR of 12.7 KJ/g, which is separately 36 J·g^−1^·K^−1^, 35.9 w/g, and 3.1 KJ/g lower than those of RL membranes. In particular, RL/30%IFR membranes have an ignition temperature of 480.5 °C while exhibiting the lowest HRR, THR, and HRC, which is separately 67 J·g^−1^·K^−1^, 68.1 w/g, and 7.1 KJ/g lower than those of RL membranes. The observation substantiates that the incorporation of IFR considerably improves RL membranes in terms of the flame safety. 

### 3.5. Effect of IFR Content on Combustion Resistance of Raw Lacquer Membranes

Table 3 exhibits the vertical combustion test (UL-94) and LOI results of RL and RL/IFR membranes. Comparatively, RL membranes have the lowest LOI of 17.9% and the combustion is drastic. The membranes are unable to put out the fire via self-extinguishing but also by generating droplets. In contrast, the incorporation of IFR significantly weakens the combustion level of RL/IFR membranes when being set alight, during which the droplets are notably absent. The more the IFR, the higher the LOI, indicating that the combustion resistance of raw lacquer membranes is significantly improved. Compared to RL membranes, RL/5%IFR and RL/10%IFR have a greater LOI by 2.4% and 3.9% with UL-94 being V-2 level. Moreover, LOI is 23.7% for RL/15%IFR and 25.4% for RL/20%IFR, both of which reach an inflammable level (22–27%). In the UL-94 test, RL/20%IFR membranes demonstrate efficient self-extinguishing, achieving V-0 level. Finally, the LOI of RL/25%IFR and RL/30%IFR is individually 27.8% and 30.2%, which reaches the flame resistance level (>27%) and is separately 9.9% and 12.3% greater than that of RL membranes. 

### 3.6. SEM Analysis 

Figure 7 shows the SEM images of surface of RL/IFR membranes as related to the IFR content. It is distinct that there are many protruding particles and a porous structure over the surface of RL/IFR membranes. In general, a porous structure is formed when the polymer constantly absorbs the moisture from matrices and when the moisture of raw lacquer is emitted. This is also the major morphology when laccase catalyzes the curing of raw lacquer [2]. In this study, the pores are caused by two factors. One is the moisture loss of RL membranes, and the other is that the poor compatibility between IFR and RL fails raw lacquer to wrap all IFR completely. In addition, the rugged surface is caused when IFR is embedded over the surface of raw lacquer. Figure 8 shows the SEM images of RL/IFR membranes as related to the IFR content. The interior of membranes is completely solidified. Because of the low compatibility between IFR and raw lacquer, there are more particles inside the membranes when IFR is increasingly added. However, the presence of IFR makes the interior structure of RL/IFR membranes more compact, which may be ascribed to the interaction between silica as well as RL’s phenolic hydroxyl that strengthens the cross-linking level of urushiol [29]. 

### 3.7. Mechanical Performances of Combustion Resistant Raw Lacquer Membranes

The curing of raw lacquer is implemented via the urushiol oxidation that is catalyzed by laccase, which means the process has a high demand on the temperature and humidity. The RL/IFR mixtures are smeared over a tinplate, and the temperature and relative humidity (RH) are specified as 25 °C and 70–80% in order to keep laccase active so as to accelerate the curing of the RL/IFR membranes. After the solidification of RL/IFR membranes, the mechanical properties, including impact resistance, flexibility, rigidity, and adhesion level are tested and the results are listed in Table 4. Table 4 shows the impact resistance and flexibility of RL/IFR membranes as related to the IFR content [30]. When the IFR content is increased from 5% to 25%, the rigidity of the membranes shows a rising trend, and then reaches the maximal level of 6H with the IFR content being 25 wt%. The adhesion reaches level one when membranes contain 5 wt% of IFR, and afterwards the adhesion reaches the maximal when membranes contain 10–25 wt% of IFR. Because raw lacquer and melamine modification in the IFR system have a lower compatibility, there are many particles presented in the RL/IFR membranes, which makes the membranes vulnerable when being exerted with an external force. As a result, the impact resistance and flexibility of RL/IFR membranes are compromised slightly. The hydroxyl group that has not undergone a condensation reaction of silica and phenolic hydroxyl of raw lacquer will interact with each other, reducing the concentration of phenolic hydroxyl group and thus strengthening the cross-linking degree. As the cohesion of the coating layer is reduced, the attachment level of membranes to metallic plates is improved, so are the rigidity and adhesion. Nonetheless, when IFR is increased from 25 wt% to 30 wt%, the rigidity is decreased from 6H to 5H, and the adhesion regresses to level 1. Because within a certain content, silica has formed a complete network structure, after which increasing the silica content only leads to agglomeration that adversely affects the rigidity and adhesion of RL/IFR membranes. 

### 3.8. Optics Properties of Flame Retardant RL/IFR Membranes

Table 5 shows the luster of RL/IFR membranes and the luster level first increases and then decreases because of a rise in the IFR content. In particular, the membranes exhibit the greatest luster level when containing 20–25% of IFR. It is surmised that silicon dioxide is polymerized urushiol to a greater extent due to the silicon dioxide reacting with phenolic hydroxyl groups. Subsequently, the pore amount of the surface is reduced and at the same time brightening the luster of the RL/IFR membranes. Afterwards IFR is constantly increased, IFR is embedded over the surface of RL/IFR membranes, resulting in a rugged surface and a lower luster level. 

### 3.9. Chemical Corrosion Resistance of Combustion Resistant RL/IFR Films

Raw lacquer cross-links with reticular polymer during the curing, and the curing level can be measured according to the solvent resistance of the lacquer films [14]. After curing, lacquer films are immersed in a 10% H_2_SO_4_ solution in room temperature for seven days, examining the acid resistance. The variations are listed in Table 6 where the RL do not exhibit discoloration, wrinkles, blisters, and chips regardless of whether RL is added with different contents of IFR. The results indicate that the presence of IFR does not adversely affect the high acid resistance of the membranes.

Natural lacquer membranes have low alkali resistance, so they are immersed in a 10% NaOH solution for 12 hours, examining the alkali resistance. The variation is listed in Table 7. The natural lacquer membranes do not have discoloration, wrinkles, blisters, and chips. Moreover, the addition of 10–20% of IFR reinforces the adhesion of lacquer membranes and the IFR does not fall off from the matrices, which suggests that the incorporation of IFR improves the adhesion level of lacquer membranes [15]. 

## 4. Conclusions

This study has successfully formulated compound phosphorus-nitrogen intumescent flame retardant (IFR). After IFR is added to raw lacquer, phosphorus, nitrogen, and silicon from flame retardants can exert a synergistic effect that considerably improves the combustion resistance of raw lacquer, compensating for the disadvantage of raw lacquer for practical applications. FR-1, FR-2, and diatomite are formulated for optimization, and the optimal ratio is 56 wt%, 16 wt%, and 28 wt%, respectively, with which the flame retardant functions well and is anti-droplet. Next, RL/IFR membranes are tested as related to the IFR content accordingly. The LOI of RL/30%IFR membranes is 30.2%, passing the V-0 level in the UL-94 test, with a residual char rate that is 52.86% higher than that of raw lacquer membranes. Phosphorus compounds with a low IFR content dissociate first and generate strong acids in response to heat. Strong acid effectively accelerates the dehydration and carbonization of polyols. Since then, with the continuous increase of IFR content, strong acids have only an insignificant positive effect on the dehydration and carbonization of polyols, based on the mechanism of (C_6_H_10_O_5_)n → 6nC + 5nH_2_O. The adhesion, rigidity, and luster of RL/IFR membranes are all improved, but the acid/alkali resistance, impact resistance, and flexibility remain unchanged. Moreover, the acid resistance of flame retardant membranes is not dependent on the incorporation of IFR and does not change when immersed in a 10% H_2_SO_4_ solution for seven days. By contrast, when immersed in a 10% NaOH solution, all of RL/10%IFR, RL/15%IFR, and RL/20%IFR membranes show better adhesion, and the flame retardant membranes still attach to the tinplate without falling apart. As a result, the incorporation of the compound phosphorus-nitrogen intumescent flame retardant (IFR) with the raw lacquer helps retain the original acid resistance of raw lacquer while gaining combustion resistance that compensates for the disadvantage of raw lacquer, improving the structure of char residue, making it denser and tighter judging from the SEM images, thus further reducing the heat and smoke release.

## Data Availability

The data presented in this study are available on request from the corresponding author.

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
