# Peer review of "Synthesis of a Compound Phosphorus-Nitrogen Intumescent Flame Retardant for Applications to Raw Lacquer"

_polymers, 2021, doi:10.3390/polym13172858_

Round 1
Reviewer 1 Report
Shiu et al. provided a comprehensive study to develop efficient flame retardant based on the synergistic combinations of tri(1-melamine-2-propanol) phosphate (FR-1), ammonium phosphate (FR-2), and silica particles at a ratio of 56wt%, 16 wt 179%, and 28 wt % with raw lacquer (RL). New novel phosphorus–nitrogen–silicon-containing flame retardants (IFR) were obtained via a simple preparation and formed into combustion-resistant RL/IFR membranes. The effect of IFR on the flame-retardant properties and thermal stabilities of RL/IFR membranes was investigated by LOI tests, vertical burning tests, and the microshape calorimetric and TGA analysis. The chemical structures of RL/IFR membranes were characterized by FT-IR spectroscopy and SEM. Additionally, the mechanical properties of the membranes, including impact resistance, flexibility, rigidity, and adhesion level are evaluated. The authors concluded that RL/IFR membranes have good flame retardancy, and this novel phosphorus-nitrogen combination could improve the combustion resistance of RL. The subject matter is interesting from the point of view of its practical applications because organophosphorus and phosphorus−nitrogen flame retardants appear to be very effective, produce less toxic moiety, and smoke during combustion.
Predominantly, the manuscript is well-written and technically sound but there are few drawbacks, especially regarding data interpretation and discussion, that need to be improved before publication. The weakness of this manuscript is that the discussion reflects speculation rather than the results of actual analysis. I recommend major revision along with the point of suggestions given below.
Point 1. The description of the methodology is very vague and needs to be developed especially in section 2.2.
Point 2. To verify and characterize the chemical structure of RL/IFR membranes additional methods must be performed, e. g. 1HNMR, or 31PNMR and mass spectroscopy.
Point 3. Lines 201-202. The sentence: “…the presence of an asymmetric stretch of P-O at 880 cm-1 indicates that raw lacquer covers the absorption peak of IFR” requires further clarification. Moreover, the identified functional groups and their importance should be defined in better detail in the text.
Point 4. Line 204. Please check that the assignment of the band at 1673 cm-1 is correct in Fig. 2. The spectra of all investigated RL/IFR mixtures should be presented in Fig. 2.
Point 5. Line 228. Same as above, TG curves of all investigated RL/IFR mixtures should be presented in Fig. 3.
Point 6. Lines 220-226. Please discuss these results in better detail and by using appropriate references, especially, the sentence in lines 224-227 is needed to clarify.
Point 7. FT-IR spectra of the char residue (after combustion) for RL films with different IFR contents should be performed to prove the thermal degradation mechanism discussed in the text.
Point 8. Lines 291-292. The statement is speculative because SEM images in Fig. 6, 7, and 8 show the same porous network structure, both for RL and RL/IFR films.
Point 9. Lines 341-344. Please discuss these results by using appropriate references.
Author Response
Dear reviewer:
Thank you for taking the time to review the manuscript. We have made the following replies based on the review comments. Unfortunately, we are unable to provide HNMR and PNMR data. Therefore, we have made some amendments to the title and content. In order not to misunderstand readers, we removed the word "Novel", because many experts mentioned "Novel" would think that HNMR analysis is necessary, but due to some factors we cannot provide HNMR and PNMR data in this article, so we do A few corrections have been made, and I hope you can understand. The revised part of the manuscript is separated by red font.

Reviewer 2 Report
Thank you for the opportunity to review the article entitled Synthesis of a Novel Compound Phosphorus-Nitrogen Intumescent Flame Retardant for Applications to Raw Lacquer. The article is interesting, but contains a lot of errors and understatements. I presented my comments below.
- The method of citation is inconsistent with the requirements of the Polymers journal.
- I suggest correcting the figure 1. It appears three times "R = ...". I suggest leaving R = C15,C17, and for C15 and C17 write only colons (e.g. C15:).
- Section 2.2. The authors should provide the standards according to which the measurements were carried out.
- Section 2.3.1. The authors refer to the FT-IR spectrum but do not include it. In my opinion, the spectra of the new compounds should be in the article. Moreover, the authors wrote that the band at 3469.13 cm-1 belongs to ROH. It does not belong to R-OH but to O-H. That is the difference.
- How did the authors calculate the yield of FR-1 and FR-2 reactions? From stoichiometry? The appropriate information should be placed in the manuscript text.
- Authors should also check the entire manuscript and correct superscripts and subscripts, e.g. line 163 (cm-1 and NH2).
- Figure 5. According to what standard did the authors conduct this study? There is no mention of this in the methods section.
- Have the authors investigated the residues after combustion with the FT-IR method? This study could help confirm the assumed mechanism of the combustion process of modified RLs. I think that it should be in this article.
- Section 3.4. Heat release rate is abbreviated as HRR, not HRC. HRR and PHRR are two different parameters.
- What was the largest size of obtained samples? Personally, I think the MCC study is not entirely reliable. Ideally, samples should be tested with a cone calorimeter, but this requires a considerable size of the sample (100mm x 100mm x at least 3mm).
- Conclusion section should be better supported by research results.
- Authors should cite more references. 22 references are not much for a research paper.
- The English language of the article requires moderate corrections.
Author Response
Dear reviewer:
Thank you for taking the time to review the manuscript. Based on your valuable comments, we have made the following reply. Regarding the post-combustion FTIR analysis, we re-experimented, but the peak has almost no characteristic peaks, which is very messy. We are sorry that we cannot provide post-combustion FTIR analysis data. We have corrected the formatting part, and the revised part of the manuscript is separated by red font.

Round 2
Reviewer 1 Report
I appreciate the authors' response to all comments, which is however hardly reflected in the revised manuscript. In the present form, there are still some doubts about the overall design of the study. I recommend one more revision along with the point of suggestions given below.
Point 1. The description of the methodology is very vague and needs to be developed especially in section 2.2.
Unfortunately, I don't see any improvement in the description of this section. The previous version was at least clearer but required more details of the techniques used. The authors did not make much effort when rewriting this section, and it still needs improvement.
Point 2. To verify and characterize the chemical structure of RL/IFR membranes additional methods must be performed, e. g. 1HNMR, or 31PNMR and mass spectroscopy.
I do not understand the authors' explanation of the restrictions on posting these methods. If the HNMR and PNMR analysis were already included in the patent then references can be provided.
Point 3. Lines 201-202. The sentence: “…the presence of an asymmetric stretch of P-O at 880 cm-1 indicates that raw lacquer covers the absorption peak of IFR” requires further clarification. Moreover, the identified functional groups and their importance should be defined in better detail in the text.
Inserting a reference is still not an answer to my comment.
Point 4. Line 204. Please check that the assignment of the band at 1673 cm-1 is correct in Fig. 2.
The assignment of the band at 1673 cm-1 is still not correct in Fig. 2. The band at about 1670 cm-1 should refer to melamine C=N vibrations as described by the authors (line 194).
Point 8. Lines 291-292. The statement is speculative because SEM images in Fig. 6, 7, and 8 show the same porous network structure, both for RL and RL/IFR films.
In my opinion, SEM images do not show that an increase in the phosphorus compound has a positive effect on the formation of a dense charred layer.
Author Response
Dear reviewer, thank you for reviewing our manuscript again. We have responded based on your comments. Please refer to the attachment.

Reviewer 2 Report
The article has been significantly improved. All my comments were taken into account. I recommend publishing this article in the Polymers journal.
Author Response
Thank you very much for your valuable comments on our research work, and thank you again for your review.